# Association of Vitamin D Supplementation with Cardiovascular Events: A Systematic Review and Meta-Analysis

**DOI:** 10.3390/nu14153158

**Published:** 2022-07-30

**Authors:** Yi-Yan Pei, Yu Zhang, Xing-Chen Peng, Zhe-Ran Liu, Ping Xu, Fang Fang

**Affiliations:** 1West China Hospital, Sichuan University, Chengdu 610041, China; pei_yiyan@163.com (Y.-Y.P.); pxx2014@163.com (X.-C.P.); zeno.liu@outlook.com (Z.-R.L.); 2Affiliated Hospital of Chengdu University, Chengdu 610106, China; tnt1057@outlook.com; 3Sichuan University Library, Chengdu 610044, China; xuping1057@outlook.com

**Keywords:** vitamin D, cardiovascular events, mortality, meta-analysis

## Abstract

Background: low vitamin D status has been associated with an increased incidence of cardiovascular events. However, whether vitamin D supplementation would reduce the incidence of cardiovascular events remains unclear. Purpose: To perform a systematic review and meta-analysis of the effect of vitamin D supplementation on the mortality and incidence of cardiovascular events. Data Sources: We searched Medline, Embase, and the Cochrane Central Register of Controlled Trials from their inception until 3 May 2022. Study Selection: Two authors searched for randomized clinical trials that reported vitamin D supplementation’s effect on cardiovascular events outcomes. Data Extraction: Two authors conducted independent data extraction. Data Synthesis: We identified 41,809 reports; after exclusions, 18 trials with a total of 70,278 participants were eligible for analysis. Vitamin D supplementation was not associated with the mortality of cardiovascular events (RR 0.96, 95% CI 0.88–1.06, I^2^ = 0%), the incidence of stroke (RR 1.05, 95% CI 0.92–1.20, I^2^ = 0%), myocardial infarction (RR 0.97, 95% CI 0.87–1.09, I^2^ = 0%), total cardiovascular events (RR 0.97, 95% CI 0.91–1.04, I^2^ = 27%), or cerebrovascular events (RR 1.01, 95% CI 0.87–1.18, I^2^ = 0%). Limitation: Cardiovascular events were the secondary outcome in most trials and thus, might be selectively reported. Conclusion: In this meta-analysis of randomized clinical trials, vitamin D supplementation was not associated with a lower risk of cardiovascular events than no supplementation. These findings do not support the routine use of vitamin D supplementation in general.

## 1. Introduction

Vitamin D insufficiency is highly prevalent among elderly persons worldwide [1]. A large number of observational studies have powerfully demonstrated that low level of vitamin D can negatively affect cardiovascular health and increase the incidence of cardiovascular events and mortality [2,3]. Thus, vitamin D supplementation may be a readily available, safe, and economical modality for preventing cardiovascular events [4]. However, evidence for causality from randomized controlled trials (RCTs) of the association between vitamin D supplementation and cardiovascular events is inconclusive or negative.

Recently, four systematic reviews and meta-analyses of RCTs evaluating vitamin D supplementation and cardiovascular outcomes were published [5,6,7,8]. Three reviews [5,6,7] showed no effect on cardiovascular mortality, stroke, and myocardial infarction. However, these meta-analyses typically included vitamin D and calcium co-administered trials. The calcium side-effects on cardiovascular events are uncommon but critically important, making those reviews challenging to interpret [9,10]. In parallel, the fourth review focuses on trials that utilized vitamin D as the monotherapy and suggested insufficient evidence to support vitamin D supplementation to reduce cardiovascular events. This review included only 11,841 participants, and its accuracy was limited. Subsequently, according to available evidence, the U.S. Preventive Services Task Force stated that vitamin D supplementation did not appear effective in preventing cardiovascular disease [11]. Since the publication of these systematic reviews, two additional large RCTs [12,13] have been published, supporting the potential for better-quality evidence from meta-analyses. The present systematic review and meta-analysis aimed to update the effect of vitamin D supplementation on preventing cardiovascular events.

## 2. Methods

### 2.1. Protocol and Guidance

We did the study following the Preferred Reporting Items for Systematic Reviews and Meta-Analyses (PRISMA) guidelines [14]. The predetermined protocol was registered in the PROSPERO database (CRD42019119641). The study did not require ethical approval.

### 2.2. Eligibility Criteria

Eligible criteria were as follows: (1) studies involving participants >18 years; (2) vitamin D supplements alone at any dose. (Trials of vitamin D plus calcium (or other treatment) vs. calcium alone were considered vitamin D alone interventions); (3) studies in which placebo controls or no treatment were given to the other group; (4) reported at least one of the following outcomes of interest: cardiovascular mortality, myocardial infarction, stroke, total cardiovascular events, and cardiovascular mortality and cerebrovascular events; and (5) randomized controlled trials (RCT).

Exclusion Criteria: (1) observational studies, review articles, case reports, poster abstracts, and editorials; (2) trials of vitamin D analogs or hydroxylated vitamin D; (3) trials where all participants took vitamin D; (4) trials in pregnant or lactating women; and (5) trials of critically ill patients.

### 2.3. Information Sources and Search Strategy

The retrieval strategy was developed and implemented by a medical librarian (PX). Articles published before 3 May 2022, were included. A computerized search of Medline, Embase and the Cochrane Central Register of Controlled Trials databases was conducted. Language restrictions were not applied. Details of the retrieval strategy can be obtained in Appendix A. We screened references of critical articles and reviews for additional potentially relevant articles. For ongoing and unpublished studies, we searched relevant clinical trial registries.

### 2.4. Study Selection

After deleting duplicates, all titles and abstracts were independently screened to identify potentially relevant articles by two authors (P.Y.-Y. and L.Z.-R.). Subsequently, the two authors assessed the full text of potentially relevant studies. The final list of included trials was decided on the discussion between authors. Discrepancies were resolved by consensus or arbitrated by a third author (Z.Y.).

### 2.5. Data Collection

Two authors (P.Y.-Y. and Z.Y.) independently performed full-text assessments and extracted the following data from full articles: study characteristics, patient characteristics, and vitamin D supplementation. Discrepancies between the two authors were resolved by discussion.

### 2.6. Risk of Bias Assessment 

The risk of bias in each trial was independently assessed by two authors (P.Y.-Y. and L.Z.-R.) using the Cochrane Collaboration’s tool [15,16]. Discrepancies were resolved by consensus. 

### 2.7. Quality of Evidence

Two authors (P.Y.-Y. and L.Z.-R.) independently used the Grading of Recommendation, Assessment, Development, and Evaluation (GRADE) approach to rate the evidence quality of each outcome. Depending on the strength of evidence, it was classified as high, moderate, low, or very low [17]. The GRADE guidance used the domains of study design limitations, inconsistency, indirectness, publication bias, and imprecision in results. Discrepancies were resolved by consensus.

### 2.8. Data Synthesis 

We analyzed data from the included trials using Review Manager (RevMan, version 5.4, the Nordic Cochrane Center, the Cochrane Collaboration) and R (version 4.1.2; R Project for Statistical Computing). Meanwhile, every analysis was based on the intention-to-treat approach. The meta-analysis was conducted using random-effects models despite the level of heterogeneity. The risk ratio (RR) with 95% CI was used as the summary measure. All statistical inference tests reflected a 2-sided of *p* < 0.05. I^2^ values were calculated to evaluate variation among trials due to heterogeneity [18]. If there were more than 10 RCTs in a meta-analysis, publication bias was assessed by funnel plot techniques.

### 2.9. Trial Sequential Analysis 

To explore whether cumulative data of included trials were sufficient to evaluate the results, we conducted a trial sequential analysis [19]. The purpose of the trial sequential analysis was to preserve the overall risk of type I error by 5% while retaining the power of 80% and achieving a 15% relative risk reduction from the intervention. 

### 2.10. Subgroup Analysis

Subgroup analyses were implemented based on the number of patients (≥2000 and <2000), number of events (≥200 and <200), mean age (≥70 and <70), sex (female and both), baseline 25(OH)D (≥50 and <50), after administration 25(OH)D in the experimental group (≥75 and <75), published year (before 2017 and in or after 2017), type of vitamin D (vitamin D_3_ and vitamin D_2_), daily dose equivalent (≥2000 IU/d and <2000 IU/d), residential status (community and institution), the timing of treatment (daily and intermittently), and length of follow-up (≥3 years and <3 years).

### 2.11. Sensitivity Analyses

To assess the robustness of our statistical results, we performed a series of sensitivity analyses: (1) excluding trials that were unknown or had a high risk of bias, (2) excluding the largest trial, (3) excluding quasi-randomized or cluster-randomized trials, (4) excluding trials with a high risk of bias in each domain, (5) using random-effects models.

## 3. Results

### 3.1. Characteristics of Included Studies 

The study selection process is shown in Appendix A Appendix A. The systematic search identified 41,809 records initially, of which 18 trials [20,21,22,23,24,25,26,27,28,29,30,31,32,33,34] met inclusion criteria. Table 1 shows the characteristics of these trials included in the systematic review. In total, 70,278 participants were enrolled.

### 3.2. Quality of Evidence and Risk of Bias 

GRADE summary findings for all outcomes are shown in Table 2. Risk-of-bias assessments are described in Appendix A. Of the 18 included trials, nine were at low risk of bias, eight were at unclear risk, and one was at high risk. The Funnel plots revealed no publication bias for all five outcomes (Appendix A).

### 3.3. Cardiovascular Mortality

Of these, 9 RCTs with 63,227 participants (31,620 with vitamin D supplementation and 31,607 without vitamin D) were included in the meta-analysis for cardiovascular mortality. The supplementation of vitamin D was not associated with reductions in cardiovascular mortality compared without vitamin D (RR 0.96, 95% CI 0.88–1.06, I^2^ = 0%; Figure 1).

### 3.4. Myocardial Infarction and Stroke

Stroke was reported in 12 trials with a total of 46,093 participants. Similarly, the supplementation with vitamin D was not related to reducing stroke compared with those not supplemented (RR 1.05, 95% CI 0.92–1.20, I^2^ = 0%). For myocardial infarction, 14 studies reported 1118 events (552 with vitamin D supplementation as well as 566 without vitamin D). Supplementing with vitamin D did not reduce the risk of myocardial infarction compared with not supplementing vitamin D (RR 0.97, 95% CI 0.87–1.09, I^2^ = 0%; Figure 1). 

### 3.5. Total Cardiovascular Events and Cerebrovascular Events

A total of 39,046 participants were enrolled in seven trials reporting cardiovascular events. A similar incidence of cardiovascular events was observed between the vitamin D and no vitamin D groups (RR 0.97, 95% CI 0.91–1.04, I^2^ = 27%; Figure 1. Total cerebrovascular events were similar between the two groups (RR 1.01, 95% CI 0.87–1.18, I^2^ = 0%; Figure 1).

### 3.6. Other Analyses

Prespecified subgroup analyses of all outcomes revealed no interactions between the number of patients, number of events, mean age, sex, baseline 25(OH)D, published year, type of vitamin D, daily dose equivalent of vitamin D, the timing of treatment, residential status, and length of follow-up (Table 3). The results of the sensitivity analyses were broadly consistent with the primary analysis (Appendix A Appendix A).

Trial sequential analyses of the effects of vitamin D supplementation on cardiovascular outcomes were shown in Appendix A Appendix A. In the trial sequential analysis, the pooled sample size exceeded the calculated optimum sample size for total cardiovascular events but not for cardiovascular mortality, stroke, myocardial infarction, and cerebrovascular events. 

## 4. Discussion

In this meta-analysis of 18 RCTs with a total of 70,278, supplementation of vitamin D was not associated with reducing the risk of cardiovascular mortality, myocardial infarction, stroke, total cardiovascular events, or cerebrovascular events. No significant difference between subgroup analyses was found. Evidence of trial sequential analyses indicated that vitamin D supplementation did not decrease the relative risk of myocardial infarction and total cardiovascular events by 15%. It is unlikely that further similar trials will alter the conclusions of these trial sequential analyses. However, the evidence was not enough on the outcomes of cardiovascular mortality, stroke, myocardial infarction, and cerebrovascular events.

### 4.1. Comparison with Other Studies

Our findings were consistent with previous systematic reviews on this topic [5,6,8]. Low-quality evidence of a Cochrane review suggested no significant difference in cardiovascular mortality (RR 0.98, 95% CI 0.90–1.07). A meta-analysis by Bolland et al. showed negative findings with stroke (RR 1.00, 95% CI 0.88–1.13) and myocardial infarction (RR 1.04, 95% CI 0.91–1.17) [6]. The two systematic reviews included trials with mixed interventions of vitamin D supplementation combined with calcium supplementation, which is problematic because calcium has been proven to be associated with an increased risk of cardiovascular events [38,39,40]. In the meta-analysis by Ford et al., evidence of supplementation with vitamin D alone does not appear to protect against myocardial infarction (RR 0.95, 95% CI 0.82–1.10) or stroke (RR 1.08, 95% CI 0.91–1.29) [8].

In this study, we provided a different method compared with recent meta-analyses. Our study additionally included two large RCTs, which take up 34% of the participants in this study, providing more robust results of these relationships. Moreover, our study used different selection criteria: unlike these meta-analyses, we did not include more than ten trials of vitamin D co-administered with calcium, six trials [41,42,43,44,45,46,47] of vitamin D analogs, or hydroxylated vitamin D, and one trial by Sato et al. [48] which was retracted in 2017. Furthermore, we quantified a new outcome: total cardiovascular events.

### 4.2. Implications for Future Research

Though the available evidence did not show meaningful effects of vitamin D supplementation on cardiovascular outcomes, there have been several points for future research. First, the baseline 25(OH)D concentrations of included trials may be too high. Cardiovascular events appear when the serum 25(OH)D levels are below 37 nmol/L [4]. However, only three trials [30,32,34] involving 28,520 participants reported a mean baseline 25(OH)D level of less than 37 nmol/L. The future RCTs enrolling populations with lower baseline 25(OH)D might produce different results. Second, cardiovascular disease is a chronic disease. The follow-up time of patients included in the study is only a few years. Through the different blood sampling times after administration, we can see that the vitamin D level of blood samples will decrease with the extension of sampling time. Positive results may be obtained if the drug is continuously administered for a long time and the follow-up time is extended [28]. Third, eligible trials may be carried out in the wrong populations. Like RCTs of aspirin used to prevent cardiovascular events, the cardiovascular benefits are mainly observed in participants with a high risk of cardiovascular events, such as patients with a prior history of atherosclerotic cardiovascular disease [49]. In this study, however, most participants came from primary prevention trials. Further vitamin D trials aiming to prevent cardiovascular events should design through risk factor optimization (e.g., secondary prevention). 

### 4.3. Strengths and Limitations 

We conducted the present systematic review based on a protocol published in the PROSPERO database, which followed a rigorous methodological approach derived from the Cochrane Handbook. Additionally, we assessed the quality of evidence with GRADE and the minimum information size by using trial sequential analysis. Compared with previous reviews [5,6,8], the strengths of this study included a lower degree of heterogeneity, higher quality of evidence, and large sample size, which was adequate to assess rare outcomes, as confirmed by the trial sequential analysis. Some articles [9,10] have shown that the side effects of calcium on cardiovascular events are extremely important, Previous meta-analyses usually include trials of vitamin D and calcium combinations, so it is difficult to explain the role of vitamin D. We aim to give a clearer answer of vitamin D in the occurrence of cardiovascular disease.

We are, however, aware of several limitations. First, our study focused on published trials reporting cardiovascular events. In contrast, these results were not reported in many vitamin D supplementation studies, resulting in selective reporting bias.

Second, the relatively short duration of vitamin D supplementation and follow-up left it challenging to detect benefits in preventing cardiovascular events.

Third, the 15 included trials were heterogeneous in terms of vitamin D dose, duration of treatment, comorbid conditions, population characteristics, and most important, baseline 25(OH) D level. Outcomes definitions among trials differed depending on contemporary consensus definitions, contributing to clinical heterogeneity.

## 5. Conclusions

This meta-analysis of randomized clinical trials suggests that vitamin D supplementation is not associated with a decrease in cardiovascular events. There is no evidence that the routine use of these supplements is effective in preventing cardiovascular events.

## Figures and Tables

**Figure 1 nutrients-14-03158-f001:**
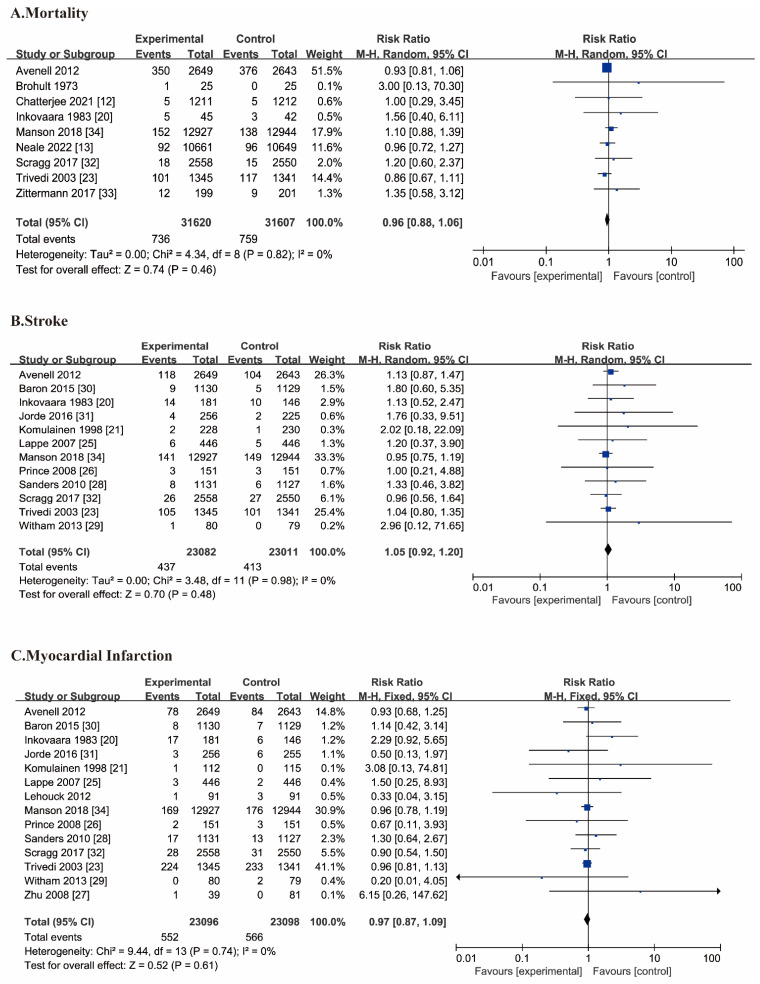
Forest plot comparing the effects of vitamin D on cardiovascular events. For different events, we presented in different rows in forest plots. A. cardiovascular mortality; B. stroke; C. myocardial infarction; D. cardiovascular events; E. cerebrovascular events. The black line represents the 95% confidence interval of each study. The blue box represents the weight of each study. The black diamond refers to results in pooled analysis [12,13,20,21,23,25,26,27,28,29,30,31,32,33,34,35,36,37].

**Table 1 nutrients-14-03158-t001:** Characteristics of the 18 Randomized Clinical Trials.

Author & Year	Mean/Median Age (SD/Range), y	Female	Participants (Vitamin D/no Vitamin D)	Baseline 25(OH)D (nmol/L) (Vitamin D/no Vitamin D)	After Administration 25(OH)D (nmol/L) (Vitamin D/no Vitamin D)	Intervention	Control	Primary Outcome	Region	Follow-Up
Brohult 1973 [35]	52 (18 to 69)	68%	25/25	NS/NS	NS/NS	vitamin D3 (100,000 IU) daily	placebo	bone mineral density	Sweden	1-year
Inkovaara 1983 [20]	79.5 (65 to 97)	17%	45/42	NS/NS	NS/NS	vitamin D3 (1000 IU) daily	placebo	Biochemistry	Tampere, Finland	1-year
Komulainen 1998 [21]	52.7 (52 to 53)	100%	228/230	NS/NS	NS/NS	vitamin D3 (300 IU) daily for four year and (100 IU) daily for the last year	placebo	bone mineral density	Kuopio, Finland	5-year
Trivedi 2003 [23]	75 (65 to 85)	24%	1345/1341	NS/NS	74/53	vitamin D3 (100,000 IU) four-monthly	placebo	Fracture	the United Kingdom	5-year
Lappe 2007 [25]	66.7 (7.3)	100%	446/445	72/72	96/71	vitamin D3 (1000 IU) plus calcium (1400 to 1500 mg) daily	calcium (1400 to 1500 mg) daily	bone mineral density	Nebraska, the USA	4-year
Prince 2008 [26]	77.2 (4.6)	100%	151/151	45/44	60/44	vitamin D2 1000 IU plus calcium 1000 mg daily	calcium (1000 mg) daily	Falls	Western Australia	1-year
Zhu 2008 [27]	74.8 (2.6)	100%	39/40	70/67	106/64	vitamin D2 (1000 IU) plus calcium (1200 mg) daily	calcium (1200 mg) daily	bone mineral density	Perth, Australia	5-year
Sanders 2010 [28]	76 (73 to 80)	100%	1131/1125	53/45	52/45	vitamin D3 (500,000 IU) yearly	placebo	falls and fractures	Victoria, Australia	3-year
Avenell 2012 [36]	77 (6)	85%	2649/2643	38/38	NS/NS	vitamin D3 (800 IU) daily	placebo	Fracture	the United Kingdom	3.75-year
Lehouck 2012 [37]	68 (9)	20%	91/91	50/50	130/54	vitamin D 100,000 IU every 4 weeks	placebo	Exacerbations in Chronic Obstructive Pulmonary Disease	Leuven, Belgium	1-year
Witham 2013 [29]	77	48%	80/79	45/45	65/45	vitamin D3 (100,000 IU) three-monthly	placebo	25OHD levels	Scotland	1-year
Baron 2015 [30]	58 (7)	37%	1130/1129	25/25	45/25	vitamin D3 (1000 IU) daily plus calcium (1200 mg) daily	calcium (1200 mg) daily	adenomas incidence	the United States	3-year
Jorde 2016 [31]	62 (9)	49%	256/255	60/61	95/59	vitamin D3 (20,000 IU) weekly	placebo	progression to type 2 diabetes	Norway	5-year
Zittermann 2017 [33]	55 (48 to 62)	27%	199/201	31/35	100/40	vitamin D3 (4000 IU) daily	placebo	all-cause mortality	Germany	3-year
Scragg 2017 [32]	NS	42%	2558/2552	64/63	135/66	vitamin D3 initial (200,000 IU) then vitamin D3 (100,000 IU) monthly	placebo	CVD and death	Auckland, New Zealand	3.3-year
Manson 2018 [34]	67.1 (7.1)	51%	12,917/12,944	30/31	41/29	vitamin D3 (2000 IU) daily	placebo	cancer and major cardiovascular events	the United States	6-year
Chatterjee 2021 [12]	60 (9.9)	44.5%	1194/1191	70/70	NS/NS	vitamin D3 4000 IU daily	placebo	cancer and major cardiovascular events	the United States	2.9-year
Neale 2022 [13]	69	45.9%	10,661/10,649	NS/NS	NS/NS	vitamin D3 60,000 IU monthly	placebo	mortality	Multiple *	5.7-year

NS, no show. * Multiple races or ethnic groups were involved.

**Table 2 nutrients-14-03158-t002:** Summary of Findings and Strength of Evidence in Studies Comparing Vitamin D vs. Control for CVD.

Outcome	No. of Patients (Studies)	Risk Ratio (95% CI)	I^2^	Absolute Effect Estimates (per 1000)			Quality
				Control	Vitamin D	Difference	
Cardiovascular mortality	41538 (6)	RR 0.96 (0.83 to 1.11)	22%	24	23	−1 (−3 to 1)	High
Stroke	46093 (12)	RR 1.05 (0.92 to 1.20)	0%	18	19	1 (−1 to 4)	High
Myocardial infarction	46184 (14)	RR 0.97 (0.87 to 1.09)	0%	25	24	−2 (−6 to 3)	High
Cardiovascular events	39046 (7)	RR 0.97 (0.91 to 1.04)	27%	68	66	−2 (−6 to 3)	High
Cerebrovascular events	39359 (10)	RR 1.01 (0.87 to 1.18)	0%	16	16	0 (−2 to 3)	High

**Table 3 nutrients-14-03158-t003:** Subgroup analysis of the effect of vitamin D supplementation for CVD.

	Cardiovascular Mortality		Stroke		Myocardial Infarction		Cardiovascular Events		Cerebrovascular Events	
Subgroup Title	RR, 95%CI	*p*	RR, 95%CI	*p*	RR, 95%CI	*p*	RR, 95%CI	*p*	RR, 95%CI	*p*
No. of patients										
≥2000	0.96 [0.87, 1.05]	0.24	1.04 [0.90, 1.19]	0.48	0.96 [0.86, 1.08]	0.65	0.97 [0.88, 1.06]	0.16	0.99 [0.84, 1.16]	0.93
<2000	1.45 [0.72, 2.93]		1.25 [0.73, 2.16]		1.12 [0.58, 2.19]		1.16 [0.91, 1.47]		1.76 [0.76, 4.10]	
No. of events										
≥200	0.95 [0.85, 1.07]	0.54	1.03 [0.89, 1.19]	0.48	0.96 [0.84, 1.09]	0.77	0.95 [0.87, 1.04]	0.04	0.99 [0.83, 1.17]	0.57
<200	1.04 [0.81, 1.32]		1.17 [0.83, 1.65]		1.00 [0.80, 1.24]		1.19 [0.97, 1.46]		1.10 [0.78, 1.56]	
Age										
≥70	0.92 [0.81, 1.03]	0.18	1.09 [0.92, 1.30]	0.44	0.99 [0.86, 1.14]	0.72	0.89 [0.81, 0.99]	0.08	1.10 [0.86, 1.41]	0.33
<70	1.06 [0.89, 1.26]		0.99 [0.80, 1.24]		0.94 [0.77, 1.15]		1.06 [0.90, 1.25]		0.94 [0.77, 1.15]	
Sex										
Female	NA	NA	1.26 [0.64, 2.48]	0.48	1.34 [0.73, 2.44]	0.29	1.15 [0.62, 2.12]	0.64	1.19 [0.52, 2.72]	0.68
Both	0.96 [0.88, 1.06]		1.04 [0.91, 1.19]		0.96 [0.85, 1.07]		0.99 [0.89, 1.10]		1.00 [0.86, 1.17]	
Baseline 25(OH)D (nmol/L)										
≥50	0.92 [0.77, 1.11]	0.56	1.06 [0.85, 1.32]	0.57	0.95 [0.81, 1.11]	0.96	0.94 [0.84, 1.06]	0.36	1.01 [0.81, 1.26]	0.73
<50	0.98 [0.86, 1.12]		1.03 [0.87, 1.22]		0.94 [0.80, 1.12]		1.04 [0.86, 1.25]		0.96 [0.76, 1.20]	
After administration 25(OH)D (nmol/L) *										
≥75	1.25 [0.74, 2.13]	0.87	1.05 [0.65, 1.68]	0.86	0.93 [0.77, 1.12]	0.34	1.07 [0.92, 1.26]	0.32	0.96 [0.57, 1.63]	0.97
<75	0.99 [0.84, 1.18]		1.01 [0.85, 1.20]		0.96 [0.83, 1.10]		0.91 [0.82, 1.01]		1.01 [0.85, 1.20]	
Published year										
Before 2017	0.92 [0.82, 1.03]	0.17	1.12 [0.94, 1.33]	0.48	0.98 [0.85, 1.12]	0.83	0.89 [0.81, 0.99]	0.04	1.12 [0.89, 1.43]	0.23
In or after 2017	1.06 [0.90, 1.26]		0.95 [0.77, 1.17]		0.95 [0.78, 1.16]		1.03 [0.94, 1.13]		0.93 [0.76, 1.14]	
Type of vitamin D										
Vitamin D3	0.96 [0.87, 1.06]	0.48	1.05 [0.92, 1.20]	0.48	0.97 [0.87, 1.08]	0.75	1.00 [0.90, 1.10]	0.76	1.01 [0.86, 1.18]	0.99
Vitamin D2	3.00 [0.13, 70.30]		1.00 [0.21, 4.88]		1.35 [0.18, 10.30]		0.83 [0.26, 2.67]		1.01 [0.27, 3.78]	
Daily dose equivalent										
≥2000 IU	1.06 [0.90, 1.26]	0.15	0.96 [0.78, 1.18]	0.48	0.93 [0.77, 1.13]	0.63	1.03 [0.94, 1.13]	0.04	0.93 [0.76, 1.14]	0.23
<2000 IU	0.92 [0.81, 1.03]		1.11 [0.94, 1.32]		0.99 [0.86, 1.14]		0.89 [0.81, 0.99]		1.12 [0.89, 1.43]	
Timing										
Daily	0.98 [0.88, 1.10]	0.56	1.05 [0.89, 1.23]	0.48	0.99 [0.84, 1.17]	0.71	1.03 [0.92, 1.15]	0.43	1.00 [0.81, 1.25]	0.95
Intermittently	0.92 [0.77, 1.11]		1.05 [0.84, 1.32]		0.95 [0.81, 1.11]		0.96 [0.83, 1.10]		1.01 [0.81, 1.26]	
Residential status										
Community	1.00 [0.87, 1.15]	0.44	1.00 [0.86, 1.17]	0.48	0.98 [0.87, 1.11]	0.60	0.99 [0.90, 1.09]	NA	0.99 [0.85, 1.16]	0.20
Institution	0.93 [0.81, 1.06]		1.18 [0.92, 1.51]		0.90 [0.68, 1.20]		NA		1.80 [0.73, 4.46]	
Follow-up										
≥3 years	0.96 [0.87, 1.06]	0.49	1.04 [0.91, 1.20]	0.48	0.96 [0.86, 1.08]	0.89	0.98 [0.89, 1.09]	0.34	0.99 [0.84, 1.16]	0.16
<3 years	1.31 [0.54, 3.16]		1.15 [0.58, 2.29]		0.89 [0.28, 2.78]		1.22 [0.79, 1.87]		1.90 [0.77, 4.70]	

NA, No available. * The mean or median level of 25(OH)D in the experimental group after administration.

## Data Availability

The datasets used and analyzed during the current study are available from the corresponding author on reasonable request.

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
