# Peer review of "Association of Vitamin D Supplementation with Cardiovascular Events: A Systematic Review and Meta-Analysis"

_nutrients, 2022, doi:10.3390/nu14153158_

Round 1
Reviewer 1 Report
This is an interesting review and meta-analysis which combines all the information from papers published from 1973 to 2022 regarding the effect of vitamin D supplementation on reducing the risk of cardiovascular events. Conclusive evidence was obtained that no protection was offered. As the authors state, the studies were very varied in the quantity of vitamin D administered and at what time intervals, the length of follow-up, the type of cardiovascular event monitored, the number of subjects, age and sex and their 25(OH)D levels prior to starting the study. However meticulous statistical methods were used to account for these differences; the same conclusion was reached in all cases that vitamin D supplementation did not lower the risk of cardiovascular events.
Some minor points for consideration by the authors follow:
1. It would be useful to add the country where each study was based to Table 1 which would give the reader some indication of ambient solar UVR exposure.
2. In addition, age might be given as a range or median in each study in Table 1.
3. If any study gives the 25(OH)D levels in the test and control groups throughout or at the end of the study period, that would be interesting to add to Table 1.
4. As is stated in line 205, cardiovascular events may only be apparent if 25(OH)D valves are below 30 ng/mL (please change this to 75 nmol/L to make it the same unit as used throughout the rest of the paper) before vitamin D supplementation. There are four studies in Table 1 (Avenell 2012, Baron 2015, Zittermann 2017, Manson 2018) in which the initial 25(OH)D levels are well below this value. Would it be worth calculating whether vitamin D supplementation had any protective effects when these studies are compared with the ones in which the initial 25(OH)D levels were considerably higher?
5. Small typographical errors;
Line 26. Change “in general people” to “in general” or “people”.
Line 31. Correct to “A large number of”
Line 37. Correct to “recently”.
Line 42. Change “the other” to “the fourth review”.
Line 56. Change “does” to “did”.
Line 68. Change “of” to “in”.
Line 106. Change “are” to “were”.
Line 112. Change “is” to “was”.
Table 1. State what NS and NR mean.
Table 3. State what NA means.
Author Response
Comment1:It would be helpful to add the country where each study was based to Table 1, which would give the reader some indication of ambient solar UVR exposure.
####Authors` Response: Thanks for your excellent suggestion. We have updated table 1 based on the information provided in the original articles.
Comment 2: In addition, age might be given as a range or median in each study in Table 1.
####Authors` Response: Thanks for your positive comment and valuable suggestion. We have updated table 1, but some articles only provide the average age and standard deviation. We have tried our best to provide more age information about patients so that readers can better understand it.
Comment 3: If any study gives the 25(OH)D levels in the test and control groups throughout or at the end of the study period, that would be interesting to add to Table 1.
####Authors` Response: Thank you again for your positive comments and valuable suggestions to improve the quality of our manuscript. We have updated table 1. According to the mean or median level of 25(OH)D in the experimental group after administration, we performed a subgroup analysis of patients and presented it in table 3.
Comment 4: As stated in line 205, cardiovascular events may only be apparent if 25(OH)D valves are below 30 ng/mL (please change this to 75 nmol/L to make it the same unit used throughout the rest of the paper) before vitamin D supplementation. There are four studies in Table 1 (Avenell 2012, Baron 2015, Zittermann 2017, Manson 2018) in which the initial 25(OH)D levels are well below this value. Would it be worth calculating whether vitamin D supplementation had any protective effects when these studies are compared with the ones in which the initial 25(OH)D levels were considerably higher?
####Authors` Response: We were sorry for our careless mistakes. Thank you for your reminder. We have checked reference 4 and corrected the errors in the article. We performed a subgroup analysis of baseline serum 25(OH)D level and post-administration 25(OH)D level. Because most cardiovascular diseases are chronic diseases, we believe that the baseline 25(OH)D level, 25(OH)D level after administration, and the long-term maintenance of the higher level of 25(OH)D in serum are of great significance to the occurrence of cardiovascular diseases.
####Change to Text: In the discussion section, Page11 line 211, Cardiovascular events appear when the serum 25(OH)D levels are below 37 nmol/L4. However, only three trials30,32,34 involving 28520 participants reported a mean baseline 25(OH)D level of less than 37 nmol/L.
####Add to Text: In the discussion section, Page11 line 214, Cardiovascular disease is a chronic disease. The follow-up time of patients included in the study is only a few years. Through the different blood sampling times after administration, we can see that the vitamin D level of blood samples will decrease with the extension of sampling time. Positive results may be obtained if the drug is continuously administered for a long time and the follow-up time is extended.
Comment 5: Small typographical errors;
Line 26. Change “in general people” to “in general” or “people”.
Line 31. Correct to “A large number of”
Line 37. Correct to “recently”.
Line 42. Change “the other” to “the fourth review”.
Line 56. Change “does” to “did”.
Line 68. Change “of” to “in”.
Line 106. Change “are” to were”.
Line 112. Change “is” to “was”.
Table 1. State what NS and NR mean.
Table 3. State what NA means.
####Authors` Response: Thanks. We have corrected these mistakes based on your suggestions.
####Change to Text: In Abstract section, Page1, line26: These findings do not support the routine use of vitamin D supplementation in general.
####Change to Text: In the Introduction section, Page1, line31: A large number of observational studies have powerfully demonstrated that low level of vitamin D can negatively affect cardiovascular health and increase the incidence of cardiovascular events and mortality.
####Change to Text: In the Introduction section, Page1, line38: Recently, four systematic reviews and meta-analyses of RCTs evaluating vitamin D supplementation and cardiovascular outcomes were published.
####Change to Text: In the Introduction section, Page1, line43: In parallel, the fourth review focuses on trials that utilized vitamin D as the monotherapy and suggested insufficient evidence to support vitamin D supplementation to reduce cardiovascular events.
####Change to Text: In the Method section, Page2, line57: The study did not require ethical approval.
####Change to Text: In the Method section, Page2, line69: (4) trials in pregnant or lactating women;
####Change to Text: In the Method section, Page3, line108: If there were more than 10 RCTs in a meta-analysis, publication bias was assessed by funnel plot techniques.
####Change to Text: In the Method section, Page2, line113: The purpose of the trial sequential analysis was to preserve the overall risk of type I error by 5% while retaining the power of 80% and achieving a 15 % relative risk reduction from the intervention.
####Add to Text: Table 1: NS, no show
####Add to Text: Table3: NA, No available

Reviewer 2 Report
In this article, the authors performed a meta analysis investigating the association of vitamin D supplementation with cardiovascular events. I think the strategies appear to be fair. However, there are a couple of concerns listed below.
There are already several meta analyses in this issue. It is not clear what are the advantages of this study, and what are implied from the differences?
When we think about the occurrence of cardiovascular events, the background, especially, the history of prior event is important. Are the studies included in this study consistent regarding this matter?
Do the authors think the amount or concentration of Vitamin D matters on this issue? Can the authors account for this matter?
Author Response
Comment1: There are already several meta-analyses in this issue. It is not clear what are the advantages of this study and what is implied from the differences?
####Authors` Response: Thank you for your positive comments about improving the quality of our manuscript. Previous large meta-analyses usually include trials of vitamin D and calcium combinations. Some articles (references 9 and 10) have shown that the side effects of calcium on cardiovascular events are uncommon but extremely important, so these reviews are challenging to explain the role of vitamin D. According to your suggestion, we have further explained our advantages in 4.3. strengths and limitations.
####Change to Text: In the discussion section, Page12 Line234: Some articles have shown that the side effects of calcium on cardiovascular events are extremely important, Previous meta-analyses usually include trials of vitamin D and calcium combinations, so it is difficult to explain the role of vitamin D. we aim to give a clearer answer of vitamin D in the occurrence of cardiovascular disease.
Comment 2: When we think about the occurrence of cardiovascular events, the background, especially the history of the prior event, is essential. Are the studies included in this study consistent regarding this matter?
####Authors` Response: Thanks for your nice comments. Limited research on vitamin D and cardiovascular events as the primary outcome. In addition, we believe that screening people with low serum 25(OH)D and a high risk of cardiovascular disease as the subjects for the RCT may obtain positive results. This will be a focus of future research.
Comment 3: Do the authors think the amount or concentration of Vitamin D matters on this issue? Can the authors account for this matter?
####Authors` Response: We feel great thanks for your professional review work on our article. We believe that the serum 25(OH)D may be closely related to the occurrence of cardiovascular disease. We performed a subgroup analysis of baseline and post-administration serum 25(OH)D levels. We believe that the baseline 25(OH)D level, 25(OH)D level after administration, and the long-term maintenance of the higher level of 25(OH)D in serum are of great significance to the occurrence of cardiovascular diseases. However, there are differences in the collection time of serum after administration and the maintenance time of high-level serum vitamin D. We believe this is a significant problem to be solved in future research.
####Change to Text: In the discussion section, Page11, line 214: Cardiovascular disease is a chronic disease. The follow-up time of patients included in the study is only a few years. Through the different blood sampling times after administration, we can see that the vitamin D level of blood samples will decrease with the extension of sampling time. Positive results may be obtained if the drug is continuously administered for a long time and the follow-up time is extended.

Round 2
Reviewer 2 Report
I have no additional comment.